# The 1816 'year without a summer' in an atmospheric reanalysis

Philip Brohan<sup>1</sup>, Gilbert P. Compo<sup>2,3</sup>, Stefan Brönnimann<sup>4</sup>, Robert J. Allan<sup>1</sup>, Renate Auchmann<sup>4</sup>, Yuri Brugnara<sup>4</sup>, Prashant D. Sardeshmukh<sup>2,3</sup>, and Jeffrey S. Whitaker<sup>3</sup>

<sup>1</sup>Met Office Hadley Centre, Exeter, EX1 3PB, UK
 <sup>2</sup>CIRES/University of Colorado, Boulder, 80309-0216, USA
 <sup>3</sup>NOAA Earth System Research Laboratory/PSD
 <sup>4</sup>Oeschger Centre, University of Bern, Bern, Switzerland

Correspondence to: Philip Brohan (philip.brohan@metoffice.gov.uk)

**Abstract.** Two hundred years ago a very cold and wet summer devastated agriculture in Europe and North America, causing widespread food shortages, unrest and suffering — the "year without a summer". This is usually blamed on the eruption of Mount Tambora, in Indonesia, the previous April, but making a link between these two events has proven difficult, as the major impacts were at

5 smaller space and time-scales than we can reconstruct with tree-ring observations and climate model simulations. Here we show that the very limited network of station barometer observations for the period is nevertheless enough to enable a dynamical atmospheric reanalysis to reconstruct the daily weather of summer 1816, over much of Europe. Adding stratospheric aerosol from the Tambora eruption to the reanalysis improves its reconstruction, explicitly linking the volcano to the weather 10 impacts.

#### 1 Introduction

The summer of 1816 saw very severe weather in Europe and eastern North America (Luterbacher and Pfister, 2015). Killing frosts destroyed crops in New England, Great Britain saw cold weather and exceptional rain, and in Central Europe there were persistent cold anomalies of 3–4°C along
with increases in cloud cover and rainfall (Auchmann et al., 2012). 1816 became popularly known as the 'year without a summer' — the bad weather caused failed harvests, famine, and civil unrest; producing 'the last great subsistence crisis in the western world' (Post, 1977). This was a weather event with major human impact, and so it's a good test case for current climate models: if something similar were to happen next year, would we be able to predict it?

A likely cause is the eruption, in April 1815, of Mount Tambora, in Sumbawa, Indonesia (Raffles and Hubbard, 1816). This was the largest volcanic eruption for hundreds of years — about twice as large as the Krakatoa eruption in 1883 (Rampino and Self, 1982), and at least three times as large as the Pinatubo eruption in 1991 (Self et al., 2004). The eruption killed some 70,000 people in its immediate vicinity, and produced a global-scale cloud of stratospheric aerosol — a substantial climate forcing (Oppenheimer, 2003).

There is no doubt that the volcano had a cooling effect: multiple proxy reconstructions indicate an annual-mean cooling, of the tropics and the northern hemisphere, of around  $0.4-0.8^{\circ}C$  (Raible et al., 2016). This is supported by comparison with the well-established effects of the modern Pinatubo eruption:  $0.4-0.8^{\circ}C$  is about twice the cooling observed after Pinatubo (Morice et al., 2012), and

30 the aerosol cloud from Tambora is believed to have produced about twice the radiative forcing as that from Pinatubo as well (though this is quite uncertain (Crowley and Unterman, 2013)). This observed large-scale cooling can be simulated by General Circulation Models (GCMs), and convincingly linked to the volcanic forcing (Shindell et al., 2004).

This well-understood cooling effect, however, is much too small to explain the observed temperature anomalies causing damage in Europe: Afternoon temperatures in Geneva were, on average,

- 3.8°C cooler than usual in summer 1816 (Auchmann et al., 2012). Understanding this *extreme* climate is not straightforward — there is some additional cooling effect that is not captured by GCM studies and proxy reconstructions. To see what happened in more detail, we have used the Twentieth-Century Reanalysis system (20CR) (Compo et al., 2011), constrained by recently-recovered station
- and ship observations of barometric pressure in 1816, to reconstruct the day-by-day circulation and atmospheric state changes throughout the year.

#### 2 Reanalysis

35

The Twentieth Century Reanalysis forms an estimate of the state of the atmosphere, over the whole world, at each point in time, by combining 56 different short term weather forecasts from the NCEP atmosphere-land model (forced by sea-surface temperature, sea-ice, atmospheric composition and stratospheric aerosol fields), with surface pressure observations, using an Ensemble Kalman Filter data assimilation system. This technique provides both an estimate of the atmospheric state every six hours, and the uncertainty of that estimate. In regions where there are enough surface pressure observations, it has successfully reconstructed the atmospheric state, hour-by-hour, back to 1870

(Compo et al., 2011).

Sea-surface temperature and sea-ice boundary condition fields are not available for 1816, so for this calculation climatological fields were used — the 1861-1890 monthly average from COBE-SST2 (Hirahara et al., 2014). Two different sets of volcanic aerosol model inputs were used: in one,

the aerosol optical depths were all set to zero (no volcanic effect), in the other best-estimate actual optical depth fields were used (Crowley and Unterman, 2013).

The surface pressure observations assimilated were taken from version 4 of the International Surface Pressure Databank (Cram et al., 2015). For the year 1816, there are very few pressure observations available for use — on average, fewer than 15 for each 6-hour reanalysis time-step. This is not enough to reconstruct the global circulation, but most of these observations are from stations in Eu-

- rope: A region bounded to the west by Armagh in Northern Ireland, to the east by Gdansk in Poland, to the south by Valencia in Spain, and to the north by Ylitornio on the border between Sweden and Finland, contains 12 stations with available observations. The information from these stations augmented by observations from occasional nearby ships is sufficient to allow the 20CR system to reconstruct the hour-by-hour changes in atmospheric circulation in this region, and to calculate
- near-surface temperature variations.

### 3 Results

The reanalysis is run globally, but only in Europe are there enough digitised pressure observations to give the reanalysis useful skill, or temperature observations to validate it; so the investigation here is restricted to Europe. The sequence and location of individual weather events (highs and lows)

is reproduced with confidence, as are the resulting temperature anomalies. Figure 1 shows weather maps for some sample times.

As 20CR assimilates no temperature information (only surface pressures) comparison of reanalysis near-surface temperatures with observations serves as validation. The station temperature observations used for this validation were taken from the targeted data rescue program reported in

- Brugnara et al. (2015). This provided digitised observations for 51 stations, but in many cases the data rescued was too limited or of too poor quality to use. Good quality temperature observations were available for 23 stations (shown in figure 1 and listed in figures 4 and 5. In spite of the small quantity of assimilated pressure observations, the reanalysis is very successful in reproducing both the spatial pattern (figure 1) and time-series (figure 2) of near-surface temperature. In particular, the
- markedly and persistently cold summer that was the damaging feature of 1816 in central Europe is well reproduced. The station temperature series are too short to make climatologies (for anomalies) and so reanalysis climatologies (interpolated to the location of each station) are used instead. The climatologies used are from 1951–80 of version 2c of 20CR. This is adequate for removing the annual cycle, but the reanalysis diurnal cycle does not correspond well with the observations (not
- shown); All the temperature results shown here have sub-daily variability removed by smoothing over a 24-hour running mean.

In Europe, summer 1816 had a modest continental-scale cooling (as shown by GCM and proxy studies) but these were combined with a circulation change that produced a localised additional

**Figure 1.** A series of atmospheric states from the reanalysis: Mean-sea-level pressure anomalies (contours: solid lines are low pressure, dashed lines high pressure), and near-surface temperature anomalies (background colours, red for warm anomalies, blue for cold anomalies). Observed station temperature anomalies are shown by coloured circles, on the same scale as the reanalysis temperatures.

cooling effect — it's the combination of the two effects that produced the extreme summer in central90 Europe (figure 3).

If the Tambora eruption caused the year without a summer, a reanalysis including the effects of volcanic aerosol forcing should be a more accurate representation of observed climate in 1816 than one without volcanic forcing, and it is: Agreement between observed station and reanalysis air temperatures is consistently improved by including the volcanic forcing (figure 4). We can also attribute the circulation change directly: 20CR estimates the mean-sea-level pressure (MSLP) change over 6 hours by running a 6-hour model forecast, and then combining that forecast with the observations using the Kalman Filter to make an analysis. If the circulation anomalies were volcanically forced,

adding the volcanic aerosol forcing to 20CR should improve the pressure forecasts, reducing the analysis increments required to match the observations. Figure 5 shows exactly this effect.

#### 100 4 Conclusions

In the satellite era, dynamical reanalysis has established itself as a vital tool for reconstructing and understanding atmospheric variability and change. More recently, sparse-input reanalyses, assimilating surface pressure only, have shown the power of the technique over longer time-scales, producing reanalysis datasets covering 100 years and more (Compo et al., 2011; Poli et al., 2016). Successfully