# Peer review of "The 1816 'year without a summer' in an atmospheric reanalysis"

_Climate of the Past, 2016_

## Referee Comment (RC1) · Anonymous Referee #1 · 21 Jul 2016

Summary/Major Points While the approach used in this paper shows the potential of atmospheric reanalyses, it makes a number of claims that are not correct. Additionally, the assessment of how good the reanalysis for 1816 is only considers recently digitized temperature, pressure and other weather data (e.g. Auchmann et al., 2012 and Brugnara et al., 2016). Other long daily temperature series appear to have been ignored. The series that have been ignored are the Central England Temperature (CET) series, which is available daily from 1772. Also ignored are the daily series from Uppsala, Stockholm and also Milan and Padua. The latter two may have been used, but the sources that developed these long series are not given. It is surprising that CET is ignored as it was used in Raible et al (2016), and most of the authors were part of this extended review of the impacts of Tambora. This included a plot of daily CET values for 1816. The temperatures from the Reanalysis should be compared with CET. CET

values for 1816 are just daily mean temperatures, but it is relatively simple to average the sub-daily reanalysis temperature to a daily average. The agreement may not be great, but there are a number of MSLP values entering that should ensure that the basic wind directions across southern Britain should be well captured. The other long daily temperature series have been available in a special issue of Climatic Change from 2002 (Camuffo and Jones, 2002) that was also published as a book. Within this special issue there are papers on each of the stations which then had long and available records (e.g. Jones and Lister, 2002 for St Petersburg). St Petersburg is probably slightly too far east, but Stockholm, Uppsala, Milan and Padua have records. These could be looked at the same way you have Geneva (i.e. the whole of 1816).

Minor Points (they become major when the ignore most of the past literature)

1. The abstract says that we cannot reconstruct smaller space scales from tree-ring observations. This is incorrect. Tree-ring reconstructions are always local in scale, where the trees grow. They are not on the daily timescale, but they will be of the growing season average. They can and have been used to look at the impacts of climate change on small areas. When spatially combined, they have been used to show that the summer of 1816 was exceptionally cold across much of the mid-to-high latitudes of the Northern Hemisphere (e.g. Briffa et al., 1998). It is trees and ice cores that have clearly shown the impacts of major volcanic eruptions on summer temperatures.

2. A better reading of Raible et al. (2016) would have shown reconstructed temperature, pressure and precipitation maps for the summer of 1816. These should be compared with those in Figure 3. These sort of reconstructions are also available for individual months as well, which again could be compared with those produced by the Reanalyses. The paper reads as though these are the first time such maps have been produced, ignoring much of the past literature.

3. In the paragraph at the end of Section 1 (lines 34-41), the authors are making the

mistake of comparing local temperature change with global averages! This has been known about for decades, since the first attempt at an NH temperature average was produced in the 19th century. There have been numerous papers that have looked at this, some in the context of volcanic impacts. Europe often seems to be affected when major explosive volcanic events occur, but not always. Other regions of the world (e.g. further east into European Russia might be less affected). A possible reference for looking at the impacts of volcanic events regionally is Jones et al (2003). This paper looks at the effect of numerous eruptions (including Tambora) on regional temperature series (from CET, Fennoscandia and Central Europe). Looking regionally is very difficult to see a signal because of the much greater signal-to-noise ratio, but an analysis for Fennoscandia would be very interesting as instrumental temperatures from the northern part of this region suggest that 1816 was a normal, and not an extremely cold summer (see e.g. Klingbjer and Moberg, 2003).

References

Briffa, K.R., Jones, P.D., Schweingruber, F.H. and Osborn, T.J., 1998: Influence of volcanic eruptions on Northern Hemisphere summer temperature over the last 600 years. Nature 393, 450-455.

Camuffo, D. and Jones, P.D., (Editors) 2002: Improved Understanding of Past Climatic Variability from Early Daily European Instrumental Sources, Kluwer Academic Publishers, Dordrecht, 392pp.

Jones, P.D. and Lister, D.H., 2002: The daily temperature record for St. Petersburg, 1743 1996. Climatic Change 53, 253 258.

Jones, P.D., Moberg, A., Osborn, T.J. and Briffa, K.R., 2003: Surface climate responses to explosive volcanic eruptions seen in long European temperature records and mid-to-high latitude tree-ring density around the Northern Hemisphere, In (A. Robock and C. Oppenheimer, Eds.) Volcanism and the Earth's Atmosphere. American Geophysical Union, Washington D.C. 239-254.
Klingbjer P, Moberg A., 2003: A composite monthly temperature record from Tornedalen in northern Sweden, 1802–2002. Int J Climatol ., 23,1465–1494.

---

## Short Comment (SC1) · 24 Jul 2016

In your manuscript you make the claim that

"... in Central Europe there were persistent cold anomalies of 3–4âŮęC along 15 with increases in cloud cover and rainfall (Auchmann et al., 2012)"

However, we have observational records for the time for Stockholm, Bologna, Milan, Praha-Klementinum, Hohenpeissenberg, Armagh, Manchester, and New Haven Connecticut. In NONE OF THEM were there any anomalies which were unusual or notable in any way during the summer of 1816.

I fear that like all weather phenomena, the "missing summer" has become hugely exaggerated over the years.

Please see my post entitled "Missing the Missing Summer" for further details. It turns out that without knowing the year, you simply cannot pick out the "Year Of The Missing Summer" by eye. There's just nothing to distinguish it from the other years.

w.

https://wattsupwiththat.com/2012/04/15/missing-the-missing-summer/

---

## Referee Comment (RC2) · Anonymous Referee #2 · 3 Aug 2016

Review of

'The 1816 'year without a summer' in an atmospheric reanalysis.

by P. Brohan et al.

Recommendation: major revisions

This manuscript presents a reanalysis of daily weather in the year 1816 using the Ensemble Kalman Filter data assimilation system already used for the NCEP 20C reanalysis. For this study daily SLP values from 12 stations in Europe and occasional ship observations are assimilated. The analysis is focused on Europe, because only there the atmospheric states in the reanalysis are constrained enough by the observations to provide reasonable skill. The reanalysis is mainly validated against temperature

observations from Geneva. It shows that the circulation anomalies over Europe and the volcanic forcing from Tambora both contribute to negative summer temperature anomalies over Europe.

There is some evidence provided that the reanalysis has skill over Europe, and showing that this can be achieved by assimilating only the low number of pressure records available at the beginning of the 19th century is a finding that in principle justifies publication. However the manuscript is clearly premature for several reasons: i) the research question is not well defined, ii) the validation is quite limited and unsystematic, iii) the results shown are incomplete, iv) there is a substantial lack of conceptual clarity with respect to the interpretation of the results, as well as v) an element of overselling the relevance of the study. Given that many of the authors are very experienced I found this a bit surprising. I thus think that major revisions followed by another round of reviews are required before the manuscript can be considered for publication.

Specific comments:

L5, the statement on spatial and temporal resolutions does not make sense as tree-ring reconstructions are local (temporal resolution is indeed a limitation) and climate simulations have a typical spatial resolution on the order of 100km, which is sufficient to investigate spatial patterns on sub-continental scale, and a temporal resolution of about 30 min. Admittedly resolution and skillfull scale are not the same, but this needs to be clarified.

L18/19, statement on predictability ('if something similar were to happen next year would we be able to predict it') is fundamentally wrong for two reasons: i.) the volcanic forcing is not predictable, and ii.) one cannot conclude from the successful capture of circulation anomalies by data assimilation, which obviously uses observations, that the circulation anomalies can be either deterministically predicted in the sense of an initial value problem, or successfully simulated as the part of the response to a forcing.

L34-41, it is not convincing to restrict the discussion to Geneva. There should be a

comprehensive review of what is known on European-wide temperature and pressure anomalies over Europe in 1816 AD from both proxy-based reconstructions as well as standard, forced GCM simulations.

L51, in the 20C reanalysis SSTs, sea ice cover and assimilated pressure are all taken from observations and are thus dynamically consistent (apart from the errors they include). The climatological SST and sea ice cover used in the reanalysis can be expected to be dynamically inconsistent with the assimilated pressure observations for 1816. I think it is unclear whether this inconsistency leads to substantial problems. A thorough analysis might help to shed light on this question but might be beyond the scope of this paper. However, the text should include at least a short discussion of this issue.

L69-71, the statement 'the sequence and location of individual weather events (highs and lows)' is not sufficiently supported by the results shown. First of all some information about the temperature observations that have been used for Fig.1 should be given. Second, the dates for Fig.1 are hard to read and are in January. Given that the focus of the paper is on summer temperature anomalies it is not consistent and not informative to show an example for winter. Moreover, selecting just a few days does not qualify as a sound validation. The revised version should include a comprehensive validation of temperature anomalies at all locations for which temperature records are available, using the entire year, including a seasonal breakdown and specific statements on the skill in summer. Potential skill measures are correlations and RMSE. The analysis should include a comparison with a suitable reference simulation without data assimilation e.g. standard forced PMIP3 simulations, or running the data assimilation system with forcing but without assimilation of pressure observations. In addition to the skill measures the timeseries should be shown for several locations, not only for Geneva (Fig.2). When calculating skill measures comments on how the ensemble is dealt with should be made, so it becomes clear whether the ensemble mean has been used or scores for individual ensemble members have been averaged.

[Figure]

L91-99, The discussion of the signal of the volcanic forcing and the interpretation of Figs. 3-5 are unlogical for at least two reasons:

i.) One should distinguish between local thermodynamic forcing and forcing of large-scale circulation anomalies. The relative contributions of the two over Europe are a priory not clear. It is in principle conceivable that the forced signal over Europe is mainly dynamical, in which case including the forcing in the simulation might not increase the skill as the forced signal would be included through the assimilation of the pressure observations. This seems not to be the case, but nevertheless the possibilities need to be discussed in a conceptually sound way to guide the analysis of the results.

ii.) It is also a possibility that the temperature anomalies over Europe are a combination of a thermodynamic response to the volcanic forcing and of a temperature response to random, unforced pressure anomalies. Even in this case on might get smaller analysis increments if the volcanic forcing is included, as the individual forecasts of chaotic, quasi-random variability can be expected to be better if an atmosphere with more realistic radiative properties is used.

The circulation anomalies for summer should be shown, if the authors come to the conclusion that they might be due to the volcanic forcing the arguments need to be made clear, a comparison with the forced circulation response in PMIP3 simulations should be made, and potential forcing mechanisms that might lead to this circulation anomaly should be discussed.

L114-117, this paragraph mixes the question of prediction (see comment on L18/19) with the question of impact modelling.

L118-125, The attribution question is interesting but the discussion is not clear. This paragraph should be either deleted or improved such that the line of argument is explained more precisely.

---

## Referee Comment (RC3) · Anonymous Referee #3 · 11 Aug 2016

In this study, the approach of the 20th Century Reanalysis is used to reconstruct the dramatic events surrounding the 'year without a summer'. A set of surface pressure observations is used to constrain the weather model to reconstruct observed atmospheric circulation using the Ensemble Kalman filter (EnKf) technique. The motivation for the study is that climate models, including the changes in atmospheric dust loadings related to the Tambora event, produce a cooling which is too small when compared to in-situ observations. This suggests that a dynamical effect must have been present as well in that period. The results of the current study clearly show the ability of this approach to capture this event. The authors conclude that the atmospheric cooling directly related to the radiative forcing is actually rather small for Europe, while the cooling related to circulation effects, advecting cold air, is much larger. This makes this study a very nice showcase for the ability of the EnKf to constrain simulations.

[Figure]

My main concern with this study is that the largest part of the paper documents the ability of the EnKf to capture the 1816 summer circulation and the dynamically consistent temperature. There is already a large and convincing body of literature that shows the merits of the approach of the 20CR. While necessary for this paper, the study needs more than this result to be ready for publication. The ingredient in this study which makes it stand-out is the attribution. Strangely, the description of this part takes only 9 lines (lines 91-99).

Another issue is that embedding within the literature on the 1816 summer or other data assimilation methods is lacking. I would have expected at least mentioning the volume edited by C.R. Harington (1992). Furthermore, the approach in this study has some parallels with other studies, like that of Rasmijn et al. (2016). The similarity is that in the current study, observed circulation of the 1816 climate with and one without the volcanic aerosol loading is used while in the Rasmijn et al. study observed circulation is simulated in the present and a future climate.

After reading the 9 lines with the attribution, the reader is left with a feeling that there is must more to discovered in these simulations. Although the increase in correlations In temperature is perhaps not too dramatic (fig. 4), the pressure over the sites with instrumental records improves convincingly. In order to gain a little better understanding, it would be nice so see how pressure upstream of Europe changes in these two simulations (with and without aerosol loading). After all, conditions in the eastern US were as bad (if not worse) in the summer of 1816 than in Europe. It is perhaps possible to identify a tropical source? Even without the volcanic loading of the Tambora explosion, you are doing quite nice already in capturing the temperature and circulation!

I can agree with the part of the conclusion (line 126-127) saying "the severe weather was influenced by the volcano", but the preceding part (attributing 1816 coldness to Tambora) is too strong for the preceding analysis.

other issues the authors may want to look into

[Figure]

One motivation for the study is that new barometric pressure measurements of 1816 have become available (line39-41). My guess is that data is available for a few years around 1816 (rather than just this single year). If my hunch is right, can the authors explain why they do not take the opportunity to produce a reanalysis for a longer period? The reason for focusing on this is that the 1816 summer had low temperatures, but this also holds for the period 1790-1820 which was the last cold episode in the so-called 'Litte Ice Age'. For many areas (in Europe), temperature in summer 1816 was not at record low level.

line34-36. Are the Geneva temperatures the only motivation for this analysis or is there more widespread evidence that reconstructions and GCM fail to capture the cooling? I was wondering if the choice to highlight Geneva relates to the poem of Lord Byron "Darkness" written in Geneva in 1816?

line 56-60. There are also sub-daily pressure readings available in e.g. eastern North America (Salem, Ma.) for this year. Why are these not used?

smaller issues

line 26-30. the numbers relating to the drop in temperature seem odd: they are the same while the authors argue that they are different. The sentence is not quite right.

fig. 4 & 5: I realize that Holland is a small country, but approximating Haarlem by Amsterdam is overdoing it.

Harington, C. R. (ed.) 1992. The Year without a Summer? World climate in 1816. Canadian Museum of Nature, Ottawa, Canada

Rasmijn, L. M., van der Schrier, G., Barkmeijer, J., Sterl, A. and Hazeleger, W. 2016. The extreme 2013/2014 winter in a future climate. J. Geophys. Res. (Atmospheres), doi:10.1002/2015JD023585

---

## Referee Comment (RC4) · Anonymous Referee #4 · 2 Sep 2016

The manuscript describes the reconstruction of the weather in Europe for summer 1816 based on the well-established 20CR reanalysis system using a very small set of (surface) pressure observations. The result is , according to the authors, able to reproduce the sequence and location of individual weather events with confidence over Europe. Verification of the resulting reanalysis surface temperature with independent station data provides a good correlation. It is shown that better results are obtained when the model includes a radiative forcing that could be representative for the Mount Tombora eruption in 1815.

I would like to congratulate the authors with the exciting result of a likely accurate reconstruction, given the very low amount of synoptic input data. It demonstrates the power of atmospheric reanalysis. I do have some queries though, and I recommend publication of the manuscript subject to a minor revision that addresses this comment.

[Figure]

General comments: 1) I think the authors should elaborate a bit more on the improvement achieved when including the volcanic forcing, rather than the phrase 'and it is', increased temperature correlations (figure 4) and reduced pressure increments (figure 5).

2) It is implicitly assumed that the model bias in the 20CR reanalysis system plays a minor role. In principle the model could be biased warm. For instance, the imposed SST could be too warm. That by itself could explain a reduced temperature correlation with respect to the unbiased system. And the apparent improvement of adding volcanic forcing to the biased system could actually lead to a deterioration of the 'true' unbiased system. How can the authors rule out such a possibility? What is known about the model error and what is known about the magnitude of the systematic error in the imposed SST and sea-ice products? I would strongly recommend that the authors address this point.

---

## Short Comment (SC2) · 2 Sep 2016

Referee #4, you have raised a very valid point when you say:

"1) I think the authors should elaborate a bit more on the improve- ment achieved when including the volcanic forcing, rather than the phrase 'and it is' ..."

Looking at the data, the correlation of observations and reanalysis data WITHOUT the eruption is 0.61, with a standard error of the mean of 0.22.

Using the reanalysis data WITH the volcanoes only changes the numbers very slightly, with a new mean of 0.66, a standard error of 0.25.

A couple of points. First, rather than decreasing the scatter of the results, the addition of the eruption actually INCREASED the scatter of the results.

[Figure]

Next, the statistical significance of the change is basically zero. Given the scatter of the results, the change in the mean is far, far too small to achieve statistical significance.

This means, of course, that the authors CANNOT claim that the results using the eruption are in any way preferable to those not using the eruption.

It also explains why you had to ask the question ... because the changes were meaningless. A statistical analysis would have demonstrated that, so the authors didn't bother with the analysis ...

I fear that your point alone totally invalidates the study, as the difference between with and without the volcano is far, far too small to rely on, it could very easily be just chance.

w.

———————————————

**Reanalysis with and without volcanoes**

Fig. 1.

---

## Short Comment (SC3) · 6 Sep 2016

The air temperature anomaly correlations for the 23 stations are 0.49 0.50 0.61 0.53 0.46 0.50 0.62 0.66 0.62 0.71 0.77 0.76 0.70 0.60 0.49 0.53 0.63 0.73 0.49 0.74 0.57 0.75 0.72 (without volcanic forcing) and 0.56 0.50 0.67 0.54 0.53 0.57 0.64 0.72 0.67 0.75 0.80 0.79 0.71 0.62 0.61 0.55 0.66 0.78 0.56 0.78 0.58 0.79 0.76 (with forcing)

If we take a null hypothesis that these are independent identically distributed (IID) samples with the same mean, then we could do a t-test. The t.test function in R gives:

t = -1.3992, df = 43.903, p-value = 0.1688 alternative hypothesis: true difference in means is not equal to 0 95 percent confidence interval: -0.10186122 0.01838296 sample estimates: mean of x mean of y 0.6165217 0.6582609

[Figure]

A p-value of 0.16 would not allow us to conclude with confidence that the mean had changed, which is (I think) the point you are making here.

However, the data are NOT IID. Most obviously they are not independent - they are paired samples, so we should use the paired-sample t-test. Repeating t.test with paired=TRUE gives:

t = -7.2588, df = 22, p-value = 2.851e-07 alternative hypothesis: true difference in means is not equal to 0 95 percent confidence interval: -0.05366419 -0.02981407 sample estimates: mean of the differences -0.04173913

This is very much a significant result, reflecting the low probability that all stations would increase in correlation (as seen in the data) if the data were samples from the same distribution.

This is better, but it is still not correct, as the correlations are not samples from the same distribution (not ID): the correlation at a station depends on the precision of the reanalysis, the nature of the weather, and the quality of the observations, all of which differ between stations.

It's hard to say in this case what an appropriate null hypothesis would be - what sort of differences should we expect if the forcing really made no difference. The fact that the correlation increases for all the stations is a strong signal of improvement, but we can't do a formal significance test.

---

## Author Comment (AC1) · 27 Sep 2016

The objective of this paper was to communicate two new discoveries in historical re-analysis:

1) Reanalysis of the type pioneered by 20CR has substantial skill in reconstructing weather events even when constrained by only a handful of pressure observations - a combination of targeted data rescue and reanalysis could be used to study a much wider range of problems than previously thought.

2) Reanalysis constrained only by surface pressure, combined with the perturbed-model studies used in detection and attribution work, offers a new and powerful way to do weather event attribution.

[Figure]

These results were surprising to the paper's authors, and we thought them important, as they have the potential to be widely used.

It is clear that the paper fails in its aims: the reviews show little interest in the methodological advances we describe, and they demand a great deal of additional work on the climate of 1816 - which, while valuable in itself, won't help communicate the message of the paper.

We won't publish this in Climate of the Past.